# 'Function First': how to promote physical activity and physical function in people with long-term conditions managed in primary care? A study combining realist and co-design methods

Rebecca-Jane Law ,[1] Joseph Langley ,[2] Beth Hall ,[3]
Christopher Burton ,[4] Julia Hiscock ,[1] Lynne Williams ,[5] Val Morrison ,[6]
Andrew Lemmey ,[7] Candida Lovell-Smith,[8] John Gallanders,[8]
Jennifer Kate Cooney ,[7] Nefyn Williams [9]

For numbered affiliations see end of article.

**Correspondence to**
Dr Rebecca-Jane Law;
r.law@bangor.ac.uk

## ABSTRACT

**Objectives**  To develop a taxonomy of interventions and a programme theory explaining how interventions improve physical activity and function in people with long-term conditions managed in primary care. To co-design a prototype intervention informed by the programme theory.

**Design**  Realist synthesis combining evidence from a wide range of rich and relevant literature with stakeholder views. Resulting context, mechanism and outcome statements informed co-design and knowledge mobilisation workshops with stakeholders to develop a primary care service innovation.

**Results**  A taxonomy was produced, including 13 categories of physical activity interventions for people with long-term conditions.

**Abridged realist programme theory**  Routinely addressing physical activity within consultations is dependent on a reinforcing practice culture, and targeted resources, with better coordination, will generate more opportunities to address low physical activity. The adaptation of physical activity promotion to individual needs and preferences of people with long-term conditions helps affect positive patient behaviour change. Training can improve knowledge, confidence and capability of practice staff to better promote physical activity. Engagement in any physical activity promotion programme will depend on the degree to which it makes sense to patients and professions, and is seen as trustworthy.

**Co-design**  The programme theory informed the co-design of a prototype intervention to: improve physical literacy among practice staff; describe/develop the role of a physical activity advisor who can encourage the use of local opportunities to be more active; and provide materials to support behaviour change.

**Conclusions**  Previous physical activity interventions in primary care have had limited effect. This may be because they have only partially addressed factors emerging in our programme theory. The co-designed prototype intervention aims to address all elements of this emergent theory, but needs further development and consideration alongside current schemes and contexts (including implications relevant to COVID-19), and testing in a future study. The integration of realist and co-design methods strengthened this study.

## Strengths and limitations of this study

► Co-production with stakeholders was embedded in all stages of the project to enhance the attention to context that is characteristic of a realist approach.

► A wide range of evidence was reviewed in order to search for organisational context, characteristics of individuals, and circumstances that led to the success or failure of an interventions; focusing on evidence containing rich description where possible.

► The iterative way in which the different data sources were integrated enhanced the depth and breadth of the findings.

► We co-designed a set of flexible resources that embodied the programme theory, but which could adapt to different contexts and augment existing initiatives.

► These resources need further development and refinement before they can be used in primary care consultations.

## INTRODUCTION

In 2019 in the UK, more than 18 million adults over the age of 18 years had a long-term condition (ie, 38% of the total adult population).[1] Approximately 25% of people with one long-term condition report 'problems performing usual activities', rising to over 60% in those with three or more long-term conditions.[2] As older people accumulate more long-term conditions, they become increasingly frail.[3–5] This is one of the biggest challenges facing health and social care systems.[6]

There are known benefits of physical activity in the management of long-term conditions, including improved physical and psychosocial functioning.[7–13] However, the proportion of the adult population in England and Wales that are at least moderately active is low,[14 15] and even lower in people with long-term conditions. There is an inverse association between habitual physical activity level and multi-morbidity.[16 17]

Primary care is well placed to empower individuals and communities to improve physical activity and function, because 90% of patients' interaction with the National Health Service (NHS) occurs in this setting.[18] However, primary care management of long-term conditions typically focuses on the diagnosis and management of disease, and not on increasing physical activity.

A better way for primary care to promote physical activity and reduce functional decline is needed, and is likely to involve a complex intervention. In order to understand the active ingredients of such an intervention, a method that focuses on complexity is required. A realist approach provides a contextualised, explanatory understanding of what works, for whom, in what circumstances, in what respects and over what duration.[19–21] Integrating this with co-design gives new ideas tangible form, and tests how these will work in the real world.[22]

### Objectives

The overall aim was to conduct a realist evidence synthesis, informing the development of a primary care intervention to promote physical activity and physical function for people with long-term conditions. Specific objectives were:

1. To produce a taxonomy of physical activity interventions that aim to reduce functional decline in people with long-term conditions managed in primary care.
2. To work with patients, health professionals and researchers to uncover the complexity associated with the range of physical activity interventions in primary care, and how these directly or indirectly affect the physical functioning of people with long-term conditions.
3. To identify the mechanisms through which interventions bring about functional improvements in people with long-term conditions, and the circumstances associated with how the interventions are organised and operate within different primary care contexts.
4. To understand the potential impacts of these interventions across primary care and other settings, such as secondary healthcare and social care, paying attention to the conditions that influence how they operate.
5. To co-produce an evidence-based, theory-driven explanatory account, in the form of refined programme theory to underpin and develop a new intervention through a co-design process with patients, health professionals and researchers.

## METHOD

We performed a realist synthesis of literature following established methods[19 23] to develop context, mechanism and outcome (CMO) statements with input from key stakeholders; people with long-term conditions, health professionals and our study management and advisory groups. Stakeholders gave feedback on the emerging theories based on their lived experience as someone with a long-term condition, health professional or researcher.

Co-production was embedded throughout the following five phases over an 18 month period: (1) participatory theory-building workshops; (2) extended literature review; (3) co-design; (4) interviews and theory refinement; (5) knowledge mobilisation. The process was iterative, with data sources informing each other as the synthesis progressed (figure 1). In this study, 'co-production' refers to the co-production of the whole research project with stakeholders, and 'co-design' refers to the specific activities, within the co-produced research project, which focused on designing a set of resources. The overall methods are detailed elsewhere[24 25] and a visual summary is provided in online supplemental figure 1.

### Patient and public involvement

Five public research partners were proactively engaged throughout the project and contributed to monthly study management and quarterly external project advisory group meetings. They participated in decision-making, research activities (eg, group analysis sessions), reviewing public-facing documents, authoring reports and providing feedback on findings as they emerged.

### Participants

A stakeholder analysis enabled identification and targeting of the most relevant groups for the different stages of the synthesis and co-design.[26] It included representation from people with long-term conditions, primary care professionals, allied health professionals, third-sector organisations, council-funded initiatives, social care, policy-makers, commissioners and researchers . Stakeholders were recruited through primary care patient engagement groups, health professional groups, and academic and research support networks (see online supplemental table 1 for participant characteristics). All participants gave informed consent.

### Theory-building

Two theory-building stakeholder workshops and an early scoping search of published and grey literature developed initial ideas for programme theories. We used LEGO® Serious Play® as a participatory method for the workshops to enable expression and creativity through building models and to facilitate the sharing of experiences around physical activity and physical function (for an example, see online supplemental figure 2). A preliminary list of 'if…then' statements was developed (online supplemental table 2) which informed the first co-design

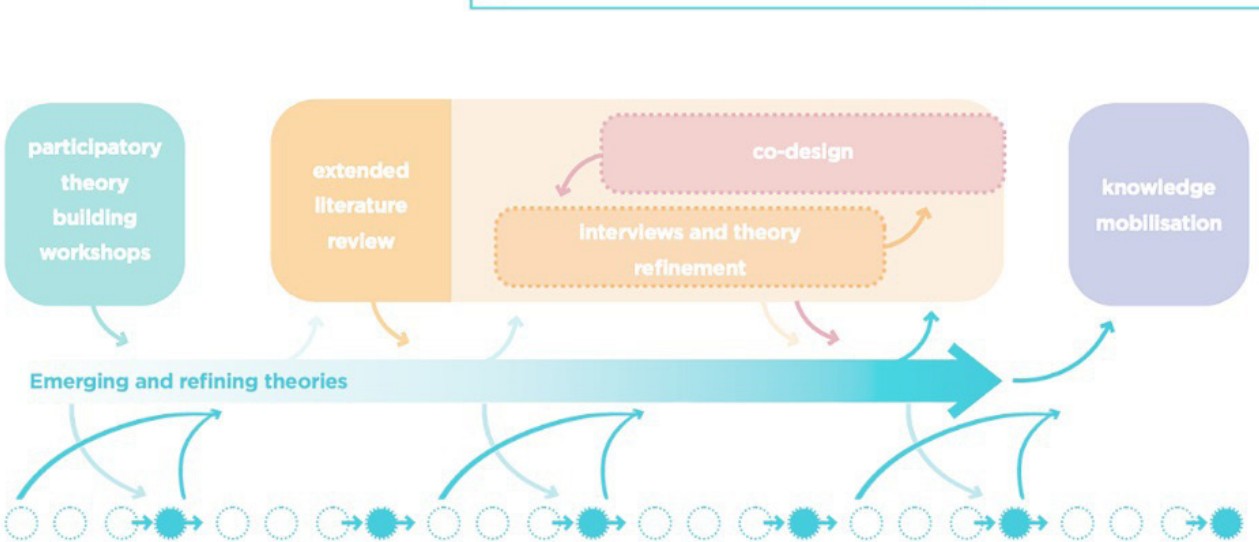

**Figure 1** Schematic showing the iterative, integrated flow of information through the following five phases over an 18-month period: (1) participatory theory-building workshops; (2) extended literature review; (3) co-design; (4) interviews and theory refinement; (5) knowledge mobilisation. Arrows indicate how each element informed another. The study management group and project advisory group meetings continuously informed the synthesis throughout the life of the project, and both groups involved input from public members.

workshop, the literature search strategy and inclusion/exclusion criteria.

### Extended literature review

We developed and amended an iterative systematic search strategy including search terms such as 'physical activity', 'physical function' and 'primary care'.[24 25] We ran searches across the bibliographic databases: Medline, CINAHL, ASSIA, Social Services Abstracts, PsycInfo and Cochrane Library. We used Covidence software[27] to coordinate the review process and apply our initial inclusion and exclusion criteria to identify potentially relevant papers (online supplemental table 3). First of all, we examined and summarised relevant systematic reviews, which informed the development of the following eight 'theory areas':

► Promoting physical literacy across the practice team;
► Framing physical activity promotion around the link between physical activity and physical function;
► Routinely assessing and promoting physical function and activity;
► Reducing time pressure by offering consultation with a credible professional;
► Linking people into existing local initiatives;
► Using behaviour change techniques;
► Tailoring advice and goals;
► Social support from others.

Our initial literature search identified 170 articles for data extraction, using bespoke data extraction forms to capture study details, findings and data relevant to the above theory areas. A total of 73 articles were selected for final inclusion because of their relevance and theoretical richness (ie, they contained explanatory information that

was detailed enough to contribute to programme theory development).[25] We supplemented the systematic search with forward and backwards citation tracking of key articles and purposive searches of guidelines, grey literature, social prescribing and physical literacy to identify 48 additional articles (figure 2). A total of 121 pieces of evidence were selected and used to develop the CMO statements (see online supplemental table 4 for final list of papers).

### Taxonomy

While reviewing the literature, we developed a taxonomy of interventions to help organise the breadth of interventions available and inform the developing programme theories. The taxonomy was added to as the project progressed.

### Interviews and theory refinement

The theory areas were explored in 'theory-refining' telephone interviews with 10 stakeholders and also as part of the first and second co-design workshops. Using the data extracted from the included papers, and through ongoing discussion within the project team and advisory group, we developed initial 'candidate' CMO statements.[25] These CMO statements were continually refined throughout the later workshops.

### Co-design

The storyboard shown in online supplemental figure 1 provides a visual representation of how the project progressed through the different stages.

Three consecutive workshops were conducted to co-design an intervention to promote physical activity for people with long-term conditions managed in primary

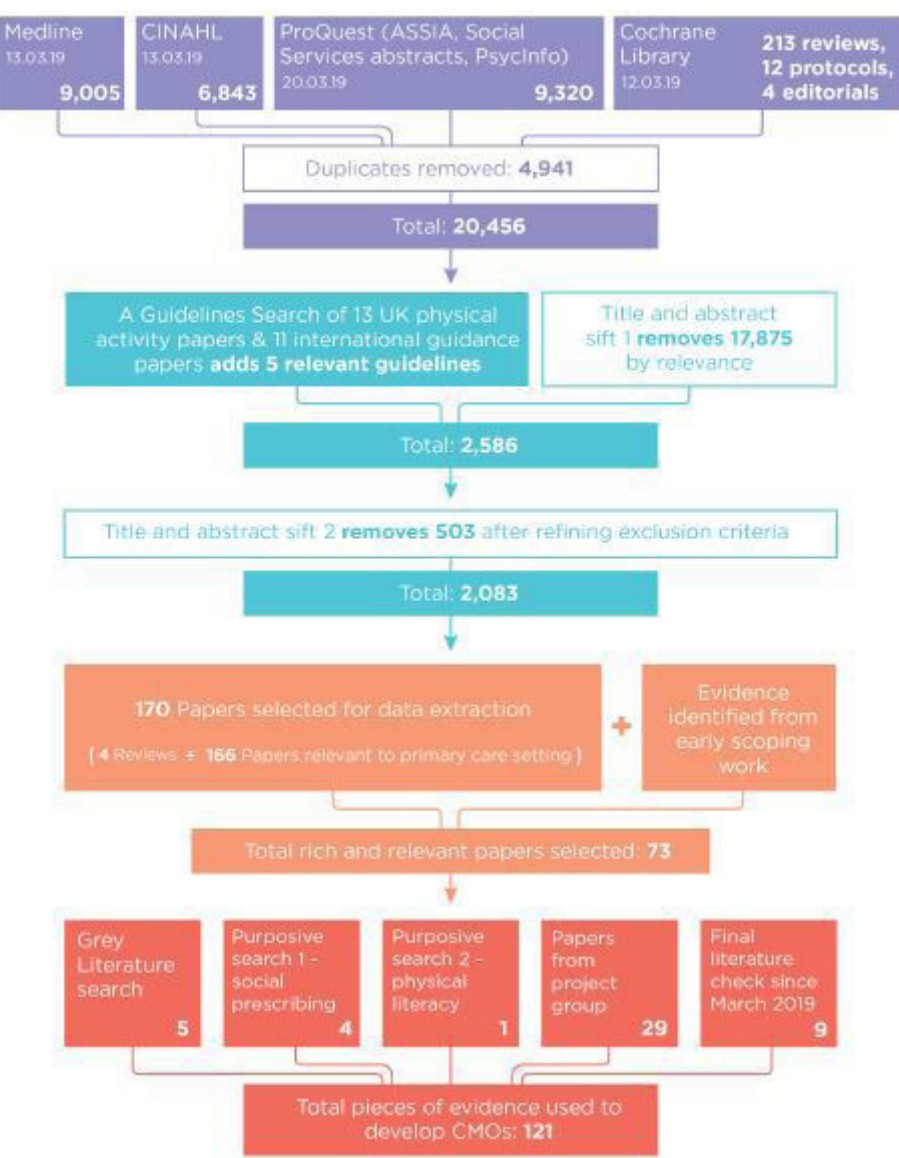

**Figure 2** Flowchart detailing the flow of information through the different phases of the review and the purposive searches.

care. The workshops were facilitated by a team of design researchers and involved a range of stakeholders (n=23) including people living with long-term conditions, primary care professionals, third sector representation, a life coach, exercise referral scheme coordinator, researchers and members of the Function First research team (see online supplemental table 1).

Using design-based activities including immersion, ideation and co-design,[25] ideas and recommendations for service innovation, and plans for making the intervention useable, were designed collaboratively and expanded during each workshop. There were key 'deliverables' from each workshop and, in between workshops, designers worked to develop ideas and provocations for the next workshop termed 'design activities'.

At the start of each workshop, the emerging programme theories and project storyboard were discussed and presented visually and verbally to inform and remind participants of the evolving context. Early indications of

theories emerging from the literature were presented to the co-design participants using card games based on the 'if…then' statements.[28] Thereafter, the relationship between the evidence and concepts was iterative; we continuously ensured that the developing CMO statements were represented and embodied in the concepts and designed products. In addition, concepts that the co-design participants raised were explored in the literature.[25]

### Knowledge mobilisation

This workshop involved people with long-term conditions, primary care professionals and researchers (n=12) and explored how best to implement the prototype intervention in different contexts, ensuring that it was desirable and feasible. The design researchers presented a physical example (or 'protoype'), which embodied the top four concepts generated in the co-design phase: ' a directory of local assets', 'a specialist role', 'training for

**Physical Variations**

**Sample Content Detail**

**Deconstruction**

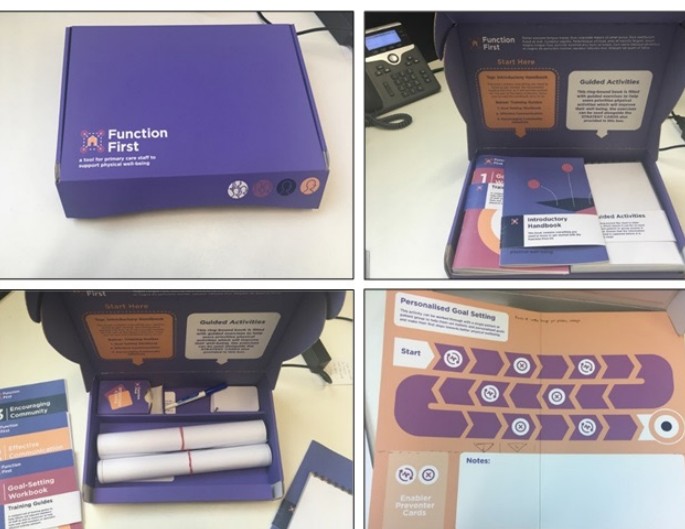

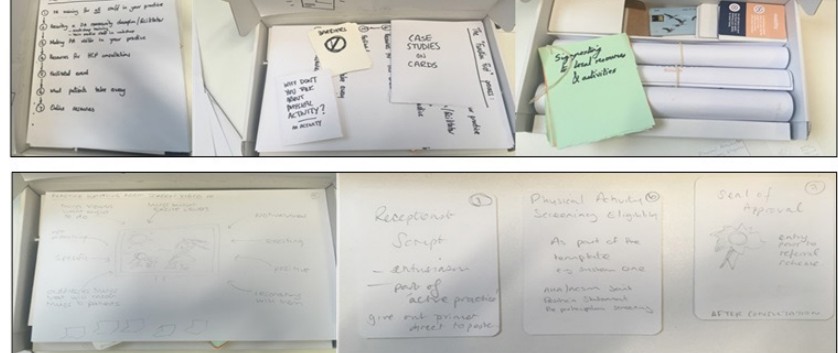

**Figure 3** Physical variations, sample content detail and an image showing how the content was deconstructed and refined as part of the workshop.

health professionals' and 'community transport'. This prototype was designed to represent and challenge these initial concepts and ideas, encourage consideration from broader perspectives, and bring together "the creativity of designers and people not trained in design together in the design development process".[29]

The co-developed ideas were refined through input from an external panel including representation from professional bodies for general practice, nursing, physiotherapy and public health. While detailed content was missing, demonstration of the intervention ideas illustrated how each physical element related to the refined CMO statements, creating an evidence-informed design solution (figure 3).

## RESULTS

A taxonomy of primary care physical activity interventions for people with long-term conditions was produced and included the following categories: brief interventions[30]; telephone interventions[31]; online/'eHealth' interventions[32]; exercise referral schemes[33]; community 'navigators'[34]; referral to exercise specialists (eg, exercise physiologists)[35]; intervention delivery by existing primary care staff[36]; physical activity 'pathways'[37–42]; practice-wide initiatives[43]; community initiatives adopted by primary care[44]; a whole system approach to embed physical activity in clinical practice[45 46]; multi-faceted interventions[47]; campaigns[48] (online supplemental table 5).

This informed the development of five CMO statements explaining how the contexts and mechanisms identified lead to outcomes relevant to improving physical activity and physical function in people with long-term conditions. Each theoretical, explanatory account below illustrates salient points with examples of evidence from the literature and stakeholder interviews.

### Changing practice culture through alignment

Programme theory: Primary care settings are characterised by competing demands, and improving physical activity and physical function is often not prioritised in a busy practice (C). If the practice team culture can be aligned to promote and support the elements of physical literacy (M), then physical activity promotion will become more routine and embedded in usual care (O).

Lack of time and competing priorities limit discussion of physical activity in primary care[49–51], as explained by a participant in this study:

> I think physical activity unfortunately does take a bit of a back step because it's probably not seen as so important as referring somebody who is expected cancer or sorting somebody's medications out. (General practitioner, individual interview)

Competing priorities include different models of care, with the primary care management of long-term conditions typically focussing on the diagnosis of disease according to the International Classification of Diseases.[52] The International Classification of Functioning, Disability and Health (ICF) places more emphasis on functional limitations in a biopsychosocial context.[53] In the context of the ICF, physical activity has the potential to promote more pro-active, 'whole person' and preventive care.[54 55] However, the time and resource limitations in primary care act as barriers to implementation of this approach.[50 56]

Physical literacy is defined as "the motivation, confidence, physical competence, knowledge and understanding to value and take responsibility for engagement in physical activities for life".[57] Aligning practice culture with physical literacy could facilitate successful physical activity promotion. A physical literacy model for adults aged 65 years and older has been developed[58] and in the UK, the Active Practice Charter aims to enhance the culture of physical activity promotion across the primary care setting.[43]

Interventions are more likely to be effective when integrated into routine practice.[59 60] For example, the 'Let's Get Moving' pathway involved embedding a physical activity promotion pathway into routine primary care practice. However, the pathway was less successful when implemented more widely, required modifications and lacked the simplicity required to align with existing programmes.[37 41 42] Care is also needed to reduce the burden of routine physical activity promotion within primary care, as explained by a participant in this study:

> But, would I want any more forms to fill in or boxes to tick or guidance that says, 'If you can touch your toes and tie up your shoelaces without getting breathless you score a one…' it wouldn't help me at all. (General practitioner, individual interview)

In order to encourage the promotion of physical activity 'as routine', protocols, pathways and procedures are insufficient; strategies are needed that align the practice team, settings and systems with the principles of physical literacy.

### Providing resources

Programme theory: Physical activity promotion in primary care is inconsistent and uncoordinated (C). If specific resources are allocated to physical activity promotion (in combination with a practice culture which is supportive) (M), then this will improve opportunities to change behaviour (O).

Despite a rise in initiatives and research,[61] physical activity promotion in primary care remains inconsistent.[62–67] Exercise referral schemes have shown small positive effects on physical activity,[33] but with low attendance and completion rates.[68 69] There are many barriers to exercise referral at an individual, social and system level.[70] To reduce burden on GPs, many interventions have allocated specific resources to physical activity promotion by identifying alternative professionals to deliver physical activity advice. Practice nurses,[71–76] healthcare assistants,[77] expert patients[78] physical activity 'coaches', 'counsellors' or 'facilitators',[79–82] exercise professionals,[83] physiotherapists,[84 85] accredited exercise physiologists[86] and different combinations of allied health professionals[54 87] have been trained to apply their existing skills and work with patients on physical activity specific goals. Furthermore, social prescribing initiatives include physical activity promotion.[88–91]

In a randomised controlled trial of referral from Australian primary care to exercise physiologists, a 12-week face-to-face and telephone coaching intervention resulted in participants completing the equivalent of 10 minutes more walking per day, which persisted after 9 months.[35] The Exercise as a Vital Sign programme delivered in the USA involved a medical assistant ascertaining a patient's self-reported physical activity prior to the GP entering the room, triggering exercise-related care processes.[92]

Primary care resource to advise patients about insufficient physical activity during routine consultations and link them to a robust referral system of physical activity opportunities could facilitate improvements in physical activity promotion and behaviour.

### Individual advice

Programme theory: People with long-term conditions have varying levels of physical function and physical activity, varying attitudes to physical activity and differing access to local resources that enable physical activity (C). If physical activity promotion is adapted to individual needs, priorities and preferences, and considers

local resource availability (M), then this will facilitate a sustained improvement in physical activity (O).

People with long-term conditions are on a spectrum of physical functioning and physical activity levels. Some people are already active, socially integrated and able to organise their everyday lives independently, whereas others have limited independence and rely on others for care.[63 93–95] People are at varying stages in the behaviour change process,[96 97] as highlighted by NICE[98] and indicated by a participant in this study:

> *There's no point in people starting to dictate to people if they're not on board with it. (Public contributor, long-term condition, individual interview)*

A variety of approaches are required to encourage people with long-term conditions to start and maintain a physically active lifestyle in a personally relevant way. The use of behaviour change techniques have been emphasised in guidance, recommending the development of goals that consider individual contexts and the impact of social support.[99 100] One-to-one sessions can be helpful to enable initial tailoring and review, whereas group-based activities can offer alternative sources of motivation.[83] Group consultations for people with long-term conditions have shown positive effects, also indicating the potential for use when resources are limited.[101 102]

Physical activity advice needs to avoid being too demanding,[103] while providing sufficient challenge.[104] Interventions have also acknowledged the unpredictable nature of living with a long-term condition by incorporating the ability to make adjustments over time.[74 75 77 102 105–108] Tailoring should link physical activity with personally relevant, enjoyable activities that are perceived as a 'good return' for the time and effort invested.[86 109–111] This could include canine-based interventions and community football schemes.[112 113] Alternative ways of providing advice include online[32 114 115] or telephone counselling,[31 116 117] which may be preferable for some people. Incorporating individualised, relevant and tailored advice has the potential to maximise relevance and effectiveness.

### Improving capability of practice workforce

Programme theory: Many primary care practice staff have a lack of knowledge and confidence to promote physical activity (C). If staff develop an improved sense of capability through education and training (M), then they will increase their engagement in physical activity promotion (O).

People with long-term conditions are familiar with primary care and typically have established trust and rapport with staff;[94 118] however, staff lack knowledge due to limited training and resources. An online survey of self-selecting GPs in England found that only 20% were familiar with the national physical activity guidelines, 26% were not familiar with any physical activity assessment tools and 55% reported that they had not undertaken any training to encourage physical activity.[62] Indeed,

only very limited medical curriculum time is devoted to physical activity and health.[119–122] Evidence has shown health professionals lack confidence, knowledge and understanding about roles and responsibilities for physical activity promotion,[123 124] and have described particular difficulties delivering motivational components such as improving self-efficacy, which are then delivered less comprehensively as a result.[36 125 126 123 124]

Interventions such as 'Movement as Medicine'[47] and 'Moving Healthcare Professionals'[45 46] have addressed this need and aim to provide more training and education for primary care health professionals. 'Moving Medicine'[46] aims to help health professionals incorporate conversations about physical activity during routine care and offers online resources relevant to patients of all ages with different long-term conditions. Improved education should increase the confidence of healthcare professionals in delivering physical activity advice.[127]

### Programme credibility

Programme theory: If a programme is credible (C), then trust and confidence in the programme will develop (M) and more patients and professionals engage with the programme (O).

Established programmes that take place in hospitals or leisure centres, and are delivered by qualified personnel (eg, cardiac rehabilitation or exercise referral schemes), have a high degree of credibility due to their association with the health service, relevant regulatory bodies and inclusion as part of NICE guidance.[99] GP referrals are often chosen as a strategy because recommendation from a known and trusted professional is felt to increase uptake.[35 94 128]

A mixed-methods review of physical activity for people with osteoarthritis found that advice was viewed as valuable if it came from a knowledgeable healthcare professional who can explain why a person should do something, tailors the advice, clearly specifies what to do and explains the benefits.[118] Active health professionals are more likely to provide better, more credible and motivating advice to their patients.[129] Credibility can also be achieved by including peer-led elements as this can increase self-efficacy among patients receiving advice, enhance empathy and improve the likelihood of realistic advice being given.[78 130 131] Understanding, tolerance, taking a genuine interest, encouragement and support were also important qualities[118], as explained by a participant in the current study:

> *It needs to be someone who is really qualified, got a good track record. They do assessments… part of the assessment is talking to people for a while, not just 5 minutes and that's it. (Public contributor, long-term condition, individual interview).*

Both professionals and patients need to feel that a programme is safe[132 133] and effective in order to engage with it.[134] Professional acceptance and implementation is

more likely if an intervention is accompanied by an evaluation that determines its effectiveness and benefit.[61]

### Intervention co-design

A prototype multi-component intervention was co-designed, embodying the five programme theories and providing resources to promote physical activity and physical function for people with long-term conditions (see figure 4 for how each CMO was embodied within the prototype resources and box 1 for components of the conceptual online resource).

The prototype consisted of:

► Resources designed to encourage a culture of physical literacy among staff and within the practice.
► Suggestions for changing the physical layout of the practice and promotional materials to create an environment that encourages physical activity.
► Materials to help develop the role of a credible professional (or 'Physical Activity Advisor') who would facilitate behaviour change during bespoke consultations with people with long-term conditions.
► Identification of community resources, which can address barriers to the uptake of physical activity, such as community transport schemes.

► Plans to develop, or adapt, an electronic directory of local physical activity opportunities, clubs and groups.

## DISCUSSION

### Summary of main findings

'Function First' is the first realist evidence synthesis with embedded co-design of physical activity promotion for people with long-term conditions managed in primary care. We developed five theoretical statements of what works, for whom and in what circumstances. From this programme theory, we co-designed flexible resources for use by a dedicated person working in primary care to promote physical activity. To our knowledge, this study is the first to use creative methods from the field of co-design to develop intervention resources that embody realist programme theories, particularly in the area of physical activity promotion for the primary care management of people with long-term conditions.

### Strengths and limitations

The realist approach offered a theory-driven explanation of the promotion of physical activity and function, paying particular attention to context (ie, settings within which interventions are placed, or pre-existing factors such as

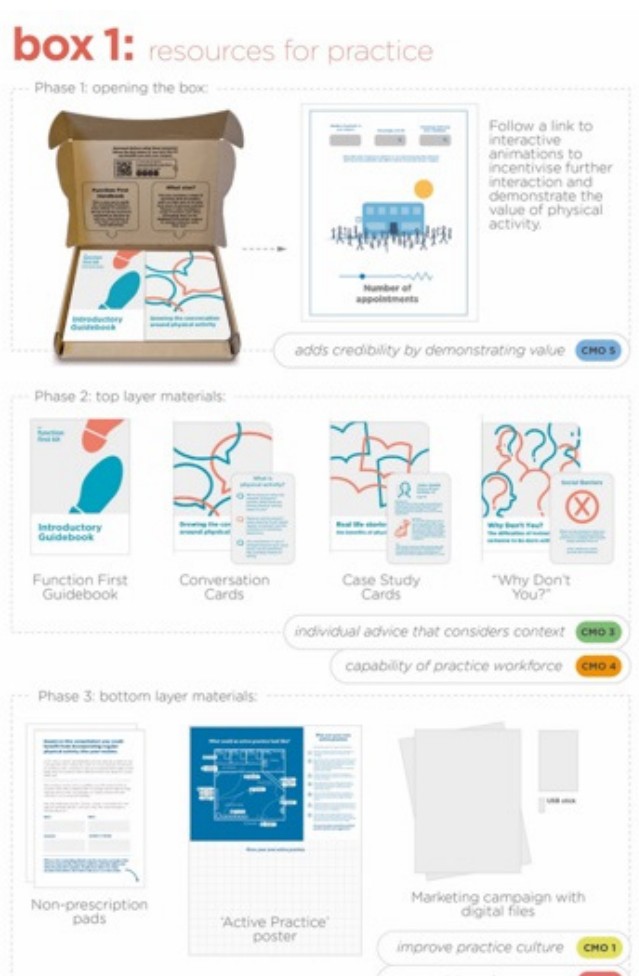

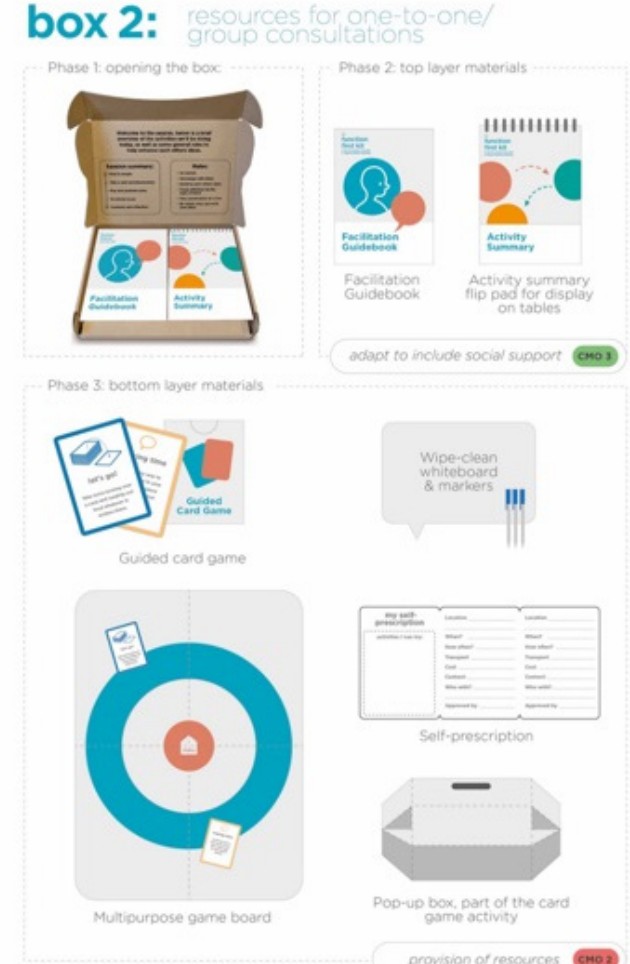

**Figure 4** Design image showing the components of boxes 1 and 2 and their relation to the CMOs.

## Box 1 Components of a conceptual online 'Function First' resource

**Patients/General Public**
Anyone participating in the Function First sessions could benefit from an online profile that tracks their progress, helps keep track of their follow-up consultation schedule and supports them with bespoke, personalised activity recommendations. The general public get access to the complete directory of local activities and transport.

**GP & Surgery**
This profile exists as a way for the GP to access the activity record of any patient attending the Function First group sessions. Each member of staff may also have a personal profile as a member of the general public to benefit from the recommendations and access to the physical activity directory.

**Credible Professional/Physical Activity Advisor (PAA)**
The Advisor could have the ability to edit the patient's profile or activity plan based on the recommendations made during a session. A part of these sessions could be a walk-through of how the online directory works. In addition to this, it would be desirable for the Advisor to begin to grow the network of activities and transport links by fostering communication between parties.

**Community Transport**
Transport services would be able to list their service in a separate transport section of the directory. Information about the operating area, capacity, number of vehicles, accessibility options and other information can be made available here, as well as direct contact information. An added benefit of this is that transport providers are often operated by volunteers who may also benefit from running this service.

**Community Activities**
Activity providers would be able to list their service in a dedicated section of the directory. Information about the activity, intensity, cost, capacity, accessibility options and other information can be made available here, as well as direct contact information. An added benefit of this is that activities are often run by volunteers who may also benefit from running this service.

motivation or organisational factors).[135] To enhance this, we embedded co-production with stakeholders at all stages, thus incorporating the different perspectives of people with long-term conditions, primary care staff, and the systems in which they live and work.

The study was planned as a linear, sequential process, but became more iterative during the course of the study. This facilitated greater integration of the different data sources and enhanced the depth and breadth of the findings.

We carried out systematic, comprehensive and transparent literature searches to identify a wide range of evidence and used Covidence software[27] to enable team contribution to reviewing the large dataset of publications. However, while we aimed to identify and present the most relevant and rich evidence, many publications lacked detailed descriptions of organisational context, characteristics of individuals, and circumstances that led to the success of the intervention. We also found fewer reports of negative results, or difficulties in implementation.

Following our stakeholder analysis, we set out to recruit people from a range of socioeconomic backgrounds, with differing ethnicity and attitudes; however, in reality this diversity proved difficult to achieve. This could be due to the timing and location of the face-to-face workshops (eg, during the day, at premises linked to the University), as well as self-selection bias whereby people supportive of and engaged with physical activity would be more likely to participate. This could be addressed in the future by offering alternative ways to participate from the outset, including remote methods[136] and dedicating more time and resource to reach out to diverse groups.

There are many initiatives promoting physical activity, and from the outset, we desired to complement rather than compete with these. Therefore, we involved representation from relevant bodies in our activities, and included a specific search for existing initiatives and campaigns.

This prototype intervention embodies all five programme theories and has been co-designed to be adaptable to different contexts. However a realist approach generates evidence-based recommendations that are related to a specific time, place and group of stakeholders and may not be applicable to alternative contexts. Similarly, co-design can be criticised for being too specific; focusing on the needs of the participants in the process, resulting in personal rather than generalisable solutions. Therefore, the current findings may not apply to a different population or set of circumstances and need further development and refinement before application.[136]

The changes to primary care associated with the COVID-19 pandemic will also need further consideration, including those related to remote consulting, practice re-organisation, use and implementation of evidence, patient behaviour and chronic disease management.[137] The need for physical activity opportunities to align with social distancing requirements and preferences, as well as mitigating against further health inequality resulting from the pandemic, will need to be considered.[138 139]

### Comparison with existing literature

Existing realist evidence syntheses within the area of physical activity promotion for people with long-term conditions have identified similar findings to the current study. For example, a realist review exploring the referral of obese adults to weight management services identified contextual factors including varying patient and practitioner characteristics and competing priorities. Practice level mechanisms included changes to systems or culture, not assuming a standardised approach, and improving communication with weight management services.[140] In addition, mechanisms proposed to maximise outcomes from exercise-based interventions for people living with chronic obstructive pulmonary disease and frailty include: trusting relationships; a shared understanding of needs; capacity to address multidimensional concerns; being

able to individualise approaches to needs and priorities; and flexible intervention delivery.[141]

Existing evidence suggests that health-related life-style advisors can remove barriers to healthy behaviour and create supportive social environments, but there is limited evidence of a positive impact on health knowledge, behaviour and outcomes.[142] The physical activity advisor role described in the current study is different to a lifestyle advisor because the role would be underpinned by knowledge and expertise specific to physical activity for people with long-term conditions.

Locating healthcare in leisure settings can create a physical environment that re-enforces physical activity culture, supports behaviour change, improves staff and patient experience, increases collaboration and coordination between health professionals, and increases awareness of facilities. Locating physical activity advisors in primary healthcare settings, as described in the current study, may have similar advantages. However, theories explaining the challenges of co-locating services highlight that the logistics of service delivery and the inconsistency of clinical schedules[143] may need further attention. In addition, theories proposed to explain what influences behaviour change practices of exercise referral practitioners, for example, may need consideration (eg, planning and training, supportive leadership, and integration between health professionals and practitioners).[144] Learning from strategies designed to combine healthcare and physical activity to create a physical activity culture across a larger population is also important(eg, 'Move More' in Sheffield).[145]

There are limited examples of applying realist methods to facilitate intervention development as conducted in the current study. In a study developing a rehabilitation intervention for elderly patients following hip fracture, three programme theories were developed: improving patient engagement by tailoring the intervention; reducing fear of falling and improving self-efficacy to exercise and perform activities of daily living; coordination of rehabilitation delivery.[146] These informed the development of an enhanced rehabilitation intervention. 'Movement as Medicine' included stakeholder work to develop a prototype intervention[47] and the 'Choose to Move' programme in Canada used participatory methods to co-create new ways to enhance physical activity, mobility and social connectedness in older adults.[147]

### Implications for practice and research

If general medical practice in the UK is to address the low levels of physical activity and poor physical functioning of people with long-term conditions, then current practice culture needs to change. A new role of a credible professional could facilitate this, with appropriate resources and protected time, increased engagement with local providers of physical activity opportunities, and full utilisation of electronic directories developed for social prescribing. Improved undergraduate and continuing medical education about physical activity is also necessary

to augment and sustain this change. The development of primary care networks, or clusters of practices, provides the opportunity for a common, shared approach. This intervention will have cost implications, but may also have direct benefits to the NHS in terms of reduced consultations and demand for services.

Addressing only some components of a programme theory may reduce the effectiveness of an intervention and explain why some existing interventions have not been successful. However the co-designed prototype intervention in this study aimed to address all components of the developed programme theory, and components of existing initiatives could also contribute to a future refined intervention.

A future planned research programme will further develop the prototype intervention, and assess its acceptability and effectiveness in the context of the Medical Research Council framework for evaluating complex interventions.[148] Remote co-design options, both digital and non-digital, that can be accessed electronically, or posted to individuals, may be needed to facilitate this development.[136] The refined intervention, resources and new role need to fit in with existing schemes (eg, National Exercise Referral Scheme and 'Moving Medicine') and complement public health campaigns (eg, 'We Are Undefeatable').[33 46 48] They also need to be flexible enough to adapt to different general medical practice contexts and changes associated with the COVID-19 pandemic. The programme theory and developed resources are relevant to the UK NHS context but could be adapted for other healthcare systems.

### CONCLUSIONS

Despite the large number of interventions promoting physical activity in primary care, physical activity levels remain low, particularly in people with long-term conditions. The limited effect of these previous interventions might be because they only partially address factors identified as important within our programme theory. The co-designed prototype intervention co-designed as part of this study addresses all elements of the programme theory, but needs further development and refinement.

**Author affiliations**
[1]North Wales Centre for Primary Care Research, Bangor University, Bangor, UK
[2]Lab4Living, Sheffield Hallam University, Sheffield, UK
[3]Library and Archives Services, Bangor University, Bangor, UK
[4]School of Allied and Public Health Professions, Canterbury Christ Church University, Canterbury, UK
[5]School of Healthcare Sciences, Bangor University, Bangor, UK
[6]School of Psychology, Bangor University, Bangor, UK
[7]School of Sport, Health and Exercise Sciences, Bangor University, Bangor, UK
[8]Patient and Public Involvement Research Partner, UK, UK
[9]Department of Primary Care and Mental Health, University of Liverpool, Liverpool, UK

Gallanders @JohnGallanders, Jennifer Kate Cooney @JenKCooney and Nefyn Williams @NefynWilliams

**Acknowledgements** We are extremely grateful to all who contributed and supported this study as stakeholders and participants through engagement in workshops, interviews and meetings across the life of the project. We would also like to acknowledge the contribution and support of the Lab4Living research team at Sheffield Hallam University: Chris Redford who created the illustrations, storyboard and the graphics for the co-designed intervention resources, all of which were imperative to the progress and outcome of this project; Rebecca Partridge who was involved in developing and facilitating the overarching creative participatory processes, including the theory building stage and the co-design stage; Remi Bec who developed and facilitated the games and participatory activities for the co-design and knowledge mobilisation workshops; and Gemma Wheeler who also help in facilitating the co-design stages. We would like to thank the following members of the study external project advisory group for their ongoing support, oversight and perspectives: Adrian Edwards (Chair), Robert Van Deursen, Julie Richardson, Asan Akpan, Jeanette Thom, Malcolm Ward, Louise Williams, Freya Davies, Andrea Hughes, Alan David Pryce. The authors would also like to thank Philip Bell who was a co-applicant and gave valuable insight from a public and patient perspective during the early stages of this project, and the research support teams within Health and Care Research Wales and the North West Coast Clinical Research Network who facilitated stakeholder involvement. Finally, from the North Wales Centre for Primary Care Research, School of Health Sciences at Bangor University, we would like to acknowledge and thank: Annie Hendry and Matthew Jones, for their help preparing the final report; Natasha Hulley, Nicola Nikolic and Richard Evans, for their ongoing administrative support throughout the project.

**Contributors** The contributions of the authors to different aspects of this work were as follows: conceiving the study and obtaining funding: R-JL, JL, BH, CB, JH, LW, VM, AL, CL-S, NW; gathering, analysis and interpretation of data: R-JL, JL, BH, CB, JH, LW, VM, AL, CL-S, JG, JC, NW; writing the report (wholly or in part): R-JL, JL, BH, CB, JH, LW, VM, AL, CL-S, JG, JC, NW; revising the report: R-JL, JL, BH, CB, JH, LW, VM, AL, CL-S, JG, JC, NW.

**Funding** This project is funded by the National Institute for Health Research (NIHR) Health Services and Delivery Research programme (17/45/22).

**Disclaimer** The views expressed are those of the author(s) and not necessarily those of the NIHR or the Department of Health and Social Care.

**Competing interests** All authors had financial support from the National Institute for Health Research (NIHR) for the submitted work; NW is a GP partner at Plas Menai Health Centre, Llanfairfechan, Wales and is member of a NIHR Health Technology Assessment programme funding committee (commissioned research); there are no other relationships or activities that could appear to have influenced the submitted work.

**Patient consent for publication** Not required.

**Ethics approval** This study received approval from the Healthcare and Medical Sciences Academic Ethics Committee, Bangor University (Reference 2018-16308) and NHS Wales Research Ethics Committee 5 (References 256729 and 262726).

**Provenance and peer review** Not commissioned; externally peer reviewed.

**Data availability statement** Data are available on reasonable request. This is an evidence synthesis involving qualitative data collection and therefore the data generated is not suitable for sharing beyond that contained within this publication and the final report published in the National Institute for Health Research journals library (Health Services and Delivery Research). Further information can be obtained from the corresponding author.

**ORCID iDs**
Rebecca-Jane Law http://orcid.org/0000-0002-1435-5086
Joseph Langley http://orcid.org/0000-0002-9770-8720
Beth Hall http://orcid.org/0000-0003-4980-3720
Christopher Burton http://orcid.org/0000-0003-1159-1494
Julia Hiscock http://orcid.org/0000-0002-8963-2981
Lynne Williams http://orcid.org/0000-0003-2575-9710
Val Morrison http://orcid.org/0000-0002-4308-8976
Andrew Lemmey http://orcid.org/0000-0003-1667-4539
Jennifer Kate Cooney http://orcid.org/0000-0002-9828-8000
Nefyn Williams http://orcid.org/0000-0002-8078-409X

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
