## [Reviewer comments · BMJ Open]

ARTICLE DETAILS

TITLE (PROVISIONAL)	'Function First': How to promote physical activity and physical function in people with long-term conditions managed in primary care? A study combining realist and co-design methods.
AUTHORS	Law, Rebecca-Jane; Langley, Joseph; Hall, Beth; Burton, Christopher; Hiscock, Julia; Williams, Lynne; Morrison, Valerie; Lemmey, Andrew; Lovell-Smith, Candida; Gallanders, John; Cooney, Jennifer; Williams, Nefyn

VERSION 1 – REVIEW

REVIEWER	Holroyd-Leduc, Jayna University of Calgary
REVIEW RETURNED	31-Jan-2021

GENERAL COMMENTS	This is an important and ambitious topic. The results provide an approach to incorporating physical activity as a focus of care within primary care. However, further edits are needed to improve the usefulness and generalizability of this publication: 1) The methods should be described in more detail, including more specifics about how the literature search was conducted and how articles were actually selected. It would also be helpful to include a summary table of the included studies.2) More details should be provided as to how exactly the literature guided the stakeholder groups, including how it was presented to participants and if/how it was used to put the findings/concepts generated at each meeting into context.3) Although the extensive supplementary materials are helpful, within the body of the manuscript It is not entirely clear how the various stakeholder groups were conducted. More details around recruitment of participants and what was actually done in each meeting should be added. How was the output from the meetings actually generated?4) Overall, the methods are not sufficiently described to enable other researchers to reproduce them. This also makes it unclear as to exactly how the authors came to the results and conclusions presented.5) The focus of the manuscript is often within the context of the UK. However, this issue is not specific to the UK. The paper would be more broadly applicable if it were rewritten with consideration of the issue of physical activity promotion within primary care more globally.
--

REVIEWER	Salmon , Victoria University of Exeter
REVIEW RETURNED	02-Feb-2021

GENERAL COMMENTS	Thank you for the opportunity to review this manuscript. It is interesting to see an example of these methods being applied to physical activity and primary care research. You have attempted to condense a lot of different data from multiple sources resulting from a complex piece of work into a single publication, which is no mean feat! My comments mainly relate to the structure of the reporting and areas for clarification, and I hope these may help with refining the manuscript: 2. is the abstract accurate, balanced and complete? The abstract could be refined and summarised further. Overall, the abstract feels a bit disconnected – a series of separate points rather than a concise summary of the project. For example, the abridged CMO statements could be summarised more concisely to allow more space for reporting the taxonomy and resources as these feel a bit ‘tagged on’. Conclusions focus on product development, but does not reflect and summarise the focus of results section. 4. Are the methods described sufficiently to be repeated? I found the methods section difficult to follow, despite the flow diagram. Although the protocol has previously been published, the whole section would benefit from clearer steps and details to allow methods to be more easily repeated. Also, consider the ordering of this section – for example, patient and public involvement is continuous throughout the project, so perhaps put this up front. It feels a bit of an afterthought at the end of the section, though clearly this wasn’t the case. Make sure the abstract and main objectives match – for example, the abstract states the objective to develop programme theory (appropriate for realist synthesis) but the method does not mention programme theory and puts the first objective to develop a taxonomy, which appears a secondary objective in the abstract. I can’t find anywhere in the methods that explains how the development of the taxonomy fits into the overall programme of work? I’d like to know more about the co-design process. You talk about co-design but I’m not sure how this worked. The illustrations in supplementary figure 1 show that Lab4Living designed the first draft and sought feedback – I’m not sure this is co-design if participants did not design it collaboratively, but just commented and refined someone else’s design? Also, you mention co-production in a few places – perhaps worth a definition of what you mean by co-production vs co-design. You may wish to look at the work of Glenn Robert regarding co-design to inform this. I wonder if it would be helpful to number the phases/workshops and present figure 1 as a timeline to help the reader keep track of what happened and when? Having some idea of timeframe will be useful to readers who might be wanting to try a similar approach. I appreciate that it was an iterative process, but I think you can still emphasis this whilst presenting a more sequential method. The story board is a great addition, but as it is not in the main text some form of ordering would be useful to help navigate the methods, as this approach is likely to be new to many readers. As you will be aware there are numerous examples of published realist syntheses/evaluations available that may help structure your manuscript a bit more clearly, for example: Abhyankar, P., Wilkinson, J., Berry, K. et al. Implementing pelvic floor muscle training for women with pelvic organ prolapse: a realist evaluation
---

	of different delivery models. BMC Health Serv Res 20, 910 (2020). https://doi.org/10.1186/s12913-020-05748-8 9. Do the results address the research question or objectives? Not clear – this relates to my previous point about the objectives and methods. You start the results with the taxonomy, in line with objective 1 of your methods section, but is this the first thing you did? Your abstract talks about programme theory and then developing a taxonomy from this? How was the taxonomy derived (see previous point in methods) and what data was used for this? It is a result of the evidence synthesis, the workshops? All of the above? 10. Are they presented clearly? See above comments. The CMO statements are well presented, but I feel that the taxonomy and co-design sections would benefit from further detail. 11. Are the discussion and conclusions justified by the results? It would be helpful to see more discussion of the results of the current synthesis and co-design process, integrating and comparing these with existing evidence, rather than presenting the existing evidence as a discreet section within the discussion. The suggestion of a role of a credible professional to facilitate physical activity is of interest, and it would be helpful to discuss how this might relate to/differ from other initiatives, for example, the Health trainer initiative. You draw the discussion to a close by suggesting implications for practice/research, but a final conclusion to summarise the overall project would enhance this section. 12. Are the study limitations discussed adequately? Limitations are discussed in some detail. It would be interesting to discuss reasons why you might not have engaged a diverse stakeholder group and how future work could address this. And one minor point - It might be helpful to mention use of Covidence software in the methods section as part of the data analysis process.
--	--

VERSION 1 – AUTHOR RESPONSE

Reviewer: 1

Dr. Jayna Holroyd-Leduc, University of Calgary Comments to the Author:

This is an important and ambitious topic. The results provide an approach to incorporating physical activity as a focus of care within primary care. However, further edits are needed to improve the usefulness and generalizability of this publication:

1) The methods should be described in more detail, including more specifics about how the literature search was conducted and how articles were actually selected. It would also be helpful to include a summary table of the included studies.

REPLY: Thank you for the opportunity to further clarify. We have provided more details about how the literature search was conducted and how articles were selected. In addition, we have uploaded a summary table of included studies as a supplementary file and referred the reader to this in the text (see page 7).

2) More details should be provided as to how exactly the literature guided the stakeholder groups, including how it was presented to participants and if/how it was used to put the findings/concepts generated at each meeting into context.

REPLY: The following article explains how the literature informed If/Then card games – which is how the early indications from the literature were presented to stakeholder groups. This has been referenced in the text. Article 323, available here: <https://research.shu.ac.uk/design4health/wp-content/uploads/2020/06/D4H-Proceedings-2020-Vol-2-Final.pdf>

As now described in the text, the relationship between the evidence and concepts was iterative; we continuously ensured that the developing CMOs were represented and embodied in the concepts and artefacts designed. In addition, concepts raised by co-design participants were explored to see whether they fitted into the literature that had already been explored, or else added new concepts. We have also added extra signposting to the storyboard which also explains this process. We hope this additional detail helps (see page 8).

3) Although the extensive supplementary materials are helpful, within the body of the manuscript It is not entirely clear how the various stakeholder groups were conducted. More details around recruitment of participants and what was actually done in each meeting should be added. How was the output from the meetings actually generated?

REPLY: Thank you for this comment. We have added further explanation of stakeholder recruitment, how the stakeholder groups were conducted and the consequent outputs were generated (see pages 7-8).

4) Overall, the methods are not sufficiently described to enable other researchers to reproduce them. This also makes it unclear as to exactly how the authors came to the results and conclusions presented.

REPLY: We welcome the opportunity to provide further detail. We have added further information and reference to the realist methods applied (see page 5), as well as further detail to explain how the stakeholder workshops were conducted (see also point 3 above).

5) The focus of the manuscript is often within the context of the UK. However, this issue is not specific to the UK. The paper would be more broadly applicable if it were rewritten with consideration of the issue of physical activity promotion within primary care more globally.

REPLY: Thank you for highlighting this. Whilst the findings of this paper are relevant to primary care health systems operating in the UK, the methods captured relevant international evidence and insights from international experts in our project advisory group. As now mentioned in the implications section of the manuscript, the programme theory and developed resources are relevant to the UK NHS context but could be adapted for other healthcare systems (see page 18).

Reviewer: 2

Dr. Victoria Salmon , University of Exeter Comments to the Author:

Thank you for the opportunity to review this manuscript. It is interesting to see an example of these methods being applied to physical activity and primary care research. You have attempted to condense a lot of different data from multiple sources resulting from a complex piece of work into a single publication, which is no mean feat! My comments mainly relate to the structure of the reporting and areas for clarification, and I hope these may help with refining the manuscript:

2. is the abstract accurate, balanced and complete?

The abstract could be refined and summarised further. Overall, the abstract feels a bit disconnected – a series of separate points rather than a concise summary of the project. For example, the abridged CMO statements could be summarised more concisely to allow more space for reporting the taxonomy and resources as these feel a bit ‘tagged on’. Conclusions focus on product development, but does not reflect and summarise the focus of results section.

REPLY: Thank you for these suggestions to improve the abstract. The sentence about the taxonomy has been moved to the beginning of the results section. The abridged CMO statements have been summarised and more clearly linked to the prototype intervention which we hope is helpful in

addressing this point. The conclusion section has been expanded to summarise the results section more fully (see page 3).

4. Are the methods described sufficiently to be repeated?

I found the methods section difficult to follow, despite the flow diagram. Although the protocol has previously been published, the whole section would benefit from clearer steps and details to allow methods to be more easily repeated.

REPLY: We have added further detail to the methods section to explain the key steps and ensure the flow diagram labels are reflected in the section headers. Within the recommended word limit there remains a need to refer back to the protocol paper however we hope the added detail helps address this concern (see pages 6-9).

Also, consider the ordering of this section – for example, patient and public involvement is continuous throughout the project, so perhaps put this up front. It feels a bit of an afterthought at the end of the section, though clearly this wasn't the case.

REPLY: Thank you for this comment. The PPI section has been moved up front as suggested (see page 6).

Make sure the abstract and main objectives match – for example, the abstract states the objective to develop programme theory (appropriate for realist synthesis) but the method does not mention programme theory and puts the first objective to develop a taxonomy, which appears a secondary objective in the abstract. I can't find anywhere in the methods that explains how the development of the taxonomy fits into the overall programme of work?

REPLY: Developing a taxonomy of interventions, which further informed the programme theory, has been added to the method section of the main text (see page 8). The order of objectives in the abstract now matches that in the main text. The method mentions developing programme theory (see pages 5-7). Thank for highlighting this, we hope these amendments help.

I'd like to know more about the co-design process. You talk about co-design but I'm not sure how this worked. The illustrations in supplementary figure 1 show that Lab4Living designed the first draft and sought feedback – I'm not sure this is co-design if participants did not design it collaboratively, but just commented and refined someone else's design? Also, you mention co-production in a few places – perhaps worth a definition of what you mean by co-production vs co-design. You may wish to look at the work of Glenn Robert regarding co-design to inform this.

REPLY: We acknowledge the concerns about the validity of the 'Co-design' process. We believe that this query may have been due to a lack of detail and perhaps design-specific phrasing and so we have added some extra explanatory detail and amended the illustration describing the process after co-design workshop 3 to include the following wording (see the storyboard in supplementary figure 1):

"Lab4Living turn the Co-design workshop ideas and specifications drawn from the research, into tangible objects and provocations that embody the design ideas and challenge them."

The prototypes the Lab4Living team made after Co-design workshop 3, were based on the concepts, ideas and specifications developed by the co-design participants. Rather than Lab4Living doing the design, the variety of prototypes were designed to represent and challenge these initial ideas, and to encourage consideration of ideas from broader perspectives. The workshop activity of developing ideas, concepts and specifications is all part of design activity. Combining this with the designerly 'making' skills and abilities of the Lab4Living team follows the definition of CoDesign as defined in the discipline of Design by Sanders & Stappers (2008):

"...the creativity of designers and people not trained in design working together in the design development process..." (Elizabeth B.-N. Sanders & Pieter Jan Stappers (2008) Co-creation and the new landscapes of design, *CoDesign*, 4:1, 5-18, DOI: 10.1080/15710880701875068)

We have added further detail to explain this (see page 9).

In this study, 'co-production' refers to the co-production of the whole research project with the project

team, advisory group, and stakeholder participants. 'Co-design' refers to a specific set of activities, within the co-produced research project, that focused on designing a set of tools/resources to enable the research findings to be put into action. We have now included a description of this in the manuscript (see page 6).

I wonder if it would be helpful to number the phases/workshops and present figure 1 as a timeline to help the reader keep track of what happened and when? Having some idea of timeframe will be useful to readers who might be wanting to try a similar approach. I appreciate that it was an iterative process, but I think you can still emphasis this whilst presenting a more sequential method. The story board is a great addition, but as it is not in the main text some form of ordering would be useful to help navigate the methods, as this approach is likely to be new to many readers. As you will be aware there are numerous examples of published realist syntheses/evaluations available that may help structure your manuscript a bit more clearly, for example: Abhyankar, P., Wilkinson, J., Berry, K. et al. Implementing pelvic floor muscle training for women with pelvic organ prolapse: a realist evaluation of different delivery models. BMC Health Serv Res 20, 910 (2020).

REPLY: Thank you for these suggestions. We have added further information to describe the phases, ordering and timescale, including more signposting to the storyboard. The elements of the methods are now numbered sequentially in the text and the wording matches that in Figure 1 (see pages 5-9). The Figure legend text is also numbered to enable easier cross-reference (see page 20).

9. Do the results address the research question or objectives?

Not clear – this relates to my previous point about the objectives and methods. You start the results with the taxonomy, in line with objective 1 of your methods section, but is this the first thing you did? Your abstract talks about programme theory and then developing a taxonomy from this? How was the taxonomy derived (see previous point in methods) and what data was used for this? It is a result of the evidence synthesis, the workshops? All of the above?

REPLY: The order of objectives in the abstract now matches that in the text. Further information about how the taxonomy was developed has been added to the methods. The taxonomy was produced whilst the programme theory was being developed. This detail is now included in the text (see page 8). Thank you for highlighting this.

10. Are they presented clearly?

See above comments. The CMO statements are well presented, but I feel that the taxonomy and co-design sections would benefit from further detail.

REPLY: Further detail has been added to describe the taxonomy and co-design elements in more detail (see pages 8 – 9).

11. Are the discussion and conclusions justified by the results?

It would be helpful to see more discussion of the results of the current synthesis and co-design process, integrating and comparing these with existing evidence, rather than presenting the existing evidence as a discreet section within the discussion. The suggestion of a role of a credible professional to facilitate physical activity is of interest, and it would be helpful to discuss how this might relate to/differ from other initiatives, for example, the Health trainer initiative.

REPLY: The 'Comparison with existing literature' and 'implications section' has been amended to include further consideration of the findings of the current study alongside those of previous studies and an explanation of how the proposed physical activity advisor role differs (see pages 16-17). We hope this improves this section, thank you for the suggestions.

You draw the discussion to a close by suggesting implications for practice/research, but a final conclusion to summarise the overall project would enhance this section.

REPLY: A short Conclusion section has been added (see page 18).

12. Are the study limitations discussed adequately?

Limitations are discussed in some detail. It would be interesting to discuss reasons why you might not have engaged a diverse stakeholder group and how future work could address this.

REPLY: We have added the following detail as requested (see pages 15-16), thank you for this comment.

Following our stakeholder analysis, we set out to recruit people with a range of socioeconomic backgrounds, ethnicity and attitudes. However we struggled to recruit a very diverse group of stakeholders. This could be due to the timing and location of the face-to-face workshops (e.g. during the day, at premises linked to the University), as well as self-selection bias whereby people supportive of and engaged with physical activity would be more likely to participate. This could be addressed in the future by offering alternative ways to participate from the outset, including remotely and dedicating more time and resources to reach out widely to more diverse groups.

And one minor point - It might be helpful to mention use of Covidence software in the methods section as part of the data analysis process.

REPLY: Covidence software has been mentioned in the methods section (see page 15).

VERSION 2 – REVIEW

REVIEWER	Salmon , Victoria University of Exeter
REVIEW RETURNED	10-May-2021

GENERAL COMMENTS	I wish to thank authors for addressing all issues in the revised article. I am happy to recommend the manuscript for publication in its current form.
---